# Cerebral Venous Thrombosis during Thyrotoxicosis: Case Report and Literature Update

**DOI:** 10.3390/jpm13111557

**Published:** 2023-10-30

**Authors:** Emanuela Maria Raho, Annibale Antonioni, Niccolò Cotta Ramusino, Dina Jubea, Daniela Gragnaniello, Paola Franceschetti, Francesco Penitenti, Andrea Daniele, Maria Chiara Zatelli, Maurizio Naccarato, Ilaria Traluci, Maura Pugliatti, Marina Padroni

**Affiliations:** 1Unit of Clinical Neurology, Neurosciences and Rehabilitation Department, University of Ferrara, 44121 Ferrara, Italy; emanuelamaria.raho@edu.unife.it (E.M.R.); annibale.antonioni@edu.unife.it (A.A.);; 2Doctoral Program in Translational Neurosciences and Neurotechnologies, University of Ferrara, 44121 Ferrara, Italy; 3Neurology Unit, Neurosciences and Rehabilitation Department, Ferrara University Hospital, 44124 Ferrara, Italy; 4Section of Endocrinology, Internal Medicine and Geriatrics, Department of Medical Sciences, University of Ferrara, 44121 Ferrara, Italy; 5Neuroradiology Unit, Ferrara University Hospital, 44124 Ferrara, Italy

**Keywords:** cerebral venous thrombosis (CVT), thyroid storm, hyperthyroidism, stroke, prothrombotic state, thyroid disease, cerebral venous sinuses, cerebral veins

## Abstract

Cerebral venous thrombosis (CVT) is a rare cause of stroke, particularly in young adults. Several known thrombophilic conditions may lead to an increased CVT risk. Interestingly, few cases in the literature have reported an association between CVT and thyrotoxicosis. Here, we describe the case of a young woman with CVT and concomitant thyrotoxicosis, without any other known prothrombotic conditions. We also performed a literature review of CVT cases and hyperthyroidism, searching for all articles published in peer-reviewed journals. We identified 39 case reports/case series concerning patients with CVT associated with thyrotoxicosis, highlighting, in most cases, the association with additional known prothrombotic factors. We then discussed the possible mechanisms by which hyperthyroidism could underlie a pro-coagulative state resulting in CVT. Thyroid disease might be a more common prothrombotic risk factor than expected in determining CVT. However, in most cases, a coexistence of multiple risk factors was observed, suggesting a multifactorial genesis of the disorder. We hope that this work may alert clinicians to consider thyrotoxicosis as a potential risk factor for CVT, even in patients who apparently have no other pro-coagulative conditions.

## 1. Introduction

Cerebral venous thrombosis (CVT) represents a rare cause of stroke, caused by blood clotting in cerebral venous sinuses and/or cerebral veins. CVT is reported to be more frequent in young females [1]. Clinical manifestations are extremely variable, depending on the site of the venous occlusion, and mainly include severe headache, acute symptomatic seizures, increase in intracranial pressure, papilledema, decreased visual acuity, oculomotor disturbances, tinnitus, focal neurological deficits, and disturbances of consciousness up to coma [2]. All thrombophilic conditions can lead to an increased CVT risk, such as hereditary coagulopathies, systemic diseases (e.g., cancer, connectivopathies, antiphospholipid antibody syndrome, or sarcoidosis), smoking, dehydration, anemia, head trauma, and head and neck infections [3]. Of note, CVT predominantly affects the female sex, particularly at a young age, and significant correlations have been shown with the use of oral contraceptives, puerperium, and pregnancy, given their tendency to promote a pro-coagulative state [4,5]. Therefore, a link between hormonal alterations and CVT has been hypothesized, and several data support this association [6,7,8]. Interestingly, few cases in the literature have reported an association between CVT and thyrotoxicosis, a clinical condition characterized by abnormally increased circulating levels of thyroid hormones [9,10]. Indeed, based on literature reports, excess circulating thyroid hormones can be associated with a hypercoagulable and hypofibrinolytic state with increased factors VIII (FVIII) and IX (FIX), fibrinogen, von Willebrand factor (vWF), and plasminogen activator inhibitor-1 (PAI-1) [11,12,13]. Therefore, since any pro-coagulative state increases the risk of CVT, a link between the two conditions has already been indicated [1]. Considering the high prevalence of thyroid diseases in the general population [14], especially in females [15], it is important to consider the possible association between CVT and thyrotoxicosis, not only to avoid potentially dangerous diagnostic delays, but also in terms of prevention, monitoring the hormonal status and avoiding the exposure to other risk factors [16]. Here, we describe the case of a young woman, with no history of endocrine diseases, who was diagnosed with CVT in the context of Graves’ disease. Furthermore, we collect similar cases in the literature, which is still rather scarce. Finally, we further explore the possible connections between these conditions, in order to help to consider thyroid diseases among the possible causes of CVT.

## 2. Case Presentation

A healthy 46-year-old woman was admitted to the emergency department of our University Hospital for severe aggravating headache and nausea that had arisen in the previous weeks. Her medical history included congenital amblyopia, a dental extraction surgery performed several years before, tonsillectomy in childhood, and chronic lower back pain. She was not taking any medications, she did not smoke, and denied any recent head trauma. At the emergency department she had blood tests, with mild D-dimer elevation, and brain computerized tomography (CT), which was unremarkable. She also underwent CT-angiography and brain magnetic resonance imaging (MRI) with venographic sequences, which found an absence of blood flow in the confluence of sinuses, ampulla of Galen, and the straight sinus, and small multifocal subacute cerebral ischemic lesions (see Figure 1 and Figure 2).

Admitted to the Neurology Unit, since the diagnostic work-up was still in progress and considering direct anticoagulants (DOACs) less explored [17], she was started on anticoagulants with low-molecular-weight heparin (LMWH) at a body weight-adjusted dose as a bridging therapy, later replaced with warfarin. The patient underwent blood tests for thrombophilic screening, namely, blood cell count, Factor V Leiden and prothrombin gene mutation, protein C, S, antithrombin III, homocysteinemia, antiphospholipid antibodies, lupus anticoagulant, angiotensin-converting enzyme (ACE), and oncomarkers (CA-19.9, CA125, CEA, NSE, and alpha-fetoprotein), which were found to be within the normal limits. Laboratory tests for any infectious etiology were negative too. Occult neoplasms and infectious outbreaks were also excluded through echocardiography and a total-body CT scan. In contrast, blood tests for the evaluation of thyroid function showed suppressed thyroid-stimulating hormone (TSH) (<0.01 μU/mL; normal values: 0.25–4.5 μU/mL) and increased triiodothyronine (T3) (19 pg/mL; normal values 2.4–4.0 pg/mL) and thyroxine (T4) levels (55.2 pg/mL; normal values: 5.5–12 pg/mL); the anti-TSH receptor antibodies were found to be positive (11 IU/L; normal values < 2.9 IU/L). Therefore, a thyroid ultrasound was performed, finding a hypervascularized and volume-increased thyroid with a non homogeneous structure. On the basis of these data, a diagnosis of thyrotoxicosis was made in the context of previously unrecognized Graves’ disease, and the patient started antithyroid therapy with 20 mg/day methimazole. Several days later, however, she developed drug-induced hepatotoxicity. Methimazole was therefore discontinued and replaced with 150 mg/day propylthiouracil with consequent progressive normalization of liver enzymes and control of hyperthyroidism. The neurological picture progressively improved, with resolution of the headache, and the clinical course was uneventful. A follow-up brain MRI documented a complete recanalization of the previously involved venous sinuses (see Figure 3). To date, the patient continues her endocrine follow-up, with good control of thyroid function with medical therapy.

## 3. Discussion

CVT, although a rare cause of brain infarction, is a potentially life-threatening condition, requiring a rapid detection and an effective treatment. Although there are several known risk factors for CVT [18,19,20], only rare cases of CVT in the context of hyperthyroidism or, generally, thyrotoxicosis, have been reported to date [21]. Moreover, only a few studies have been carried out, aimed at identifying specific pathophysiological mechanisms that may lead to thrombosis in the cerebral venous system in the context of an altered thyroid function. 

Here, we performed a literature review on the subject by querying the databases PubMed via MEDLINE, Science Direct, and Web of Science, and looking for papers (both case reports and case series) describing patients with CVT in the context of thyroid disease. We identified 39 papers (from 1996 to the present) reporting case reports or case series of individuals with CVT in the context of thyrotoxicosis (see Table 1 and Figure 4a,b for a summary of the main characteristics of the patients described in the selected papers).

Consistent with the epidemiological data on the incidence and prevalence of thyroid disorders, 66.5% of the patients described were female, and 88% of the total were aged 50 years or younger. Interestingly, a single case of CVT has been reported in an 8-year-old girl, a carrier of a mutation in heterozygosity for Leiden factor V, in the context of a thyrotoxicosis from Graves’ disease [58]. By far the most frequent (in 79% of cases) onset symptom was an aggravating headache with a progressively worsening course, accompanied by generalized tonic–clonic seizures, followed by focal neurological deficits, and altered state of consciousness. Only a few cases of deep coma were reported (5%), which required invasive neurosurgical decompression interventions due to an excessive increase in intracranial pressure [32,42]. In most cases, patients presented a range of symptoms that could be related to CVT and/or thyrotoxicosis, such as tachycardia, heat intolerance, weight loss, nausea and vomiting, and diarrhea. In the majority of cases (86%), a diagnosis of Graves’ disease was made, based on biochemical and ultrasonographic findings. Interestingly, only one case of postpartum thyroiditis [10] and one case of Hashimoto’s thyroiditis [47] were reported. The non-autoimmune forms were more frequent in males. Regarding the site of thrombosis, the most frequent site was the SSS (51%), followed by a combination of the remaining venous sinuses. In general, the SSS is known to be the most frequently involved in thrombosis [59], given its particular structure and anatomical location [60], followed by the transverse and the sigmoid sinuses [61]. The involvement of more than one venous sinus in the context of a thrombosis is also often frequent [62]. Regarding the type of treatment, most patients underwent anticoagulation with LMWH or unfractionated heparin (81.4%) as a bridge therapy, later replaced with warfarin or DOACs. Only two cases of locoregional treatment with urokinase were reported [38,43]. In 81.4% of cases, the outcome was favorable for all patients with complete recovery or only minimal disability. This proportion seems in line with those reported in the literature, regardless of CVT etiology. Indeed, evaluating various meta-analyses, Bousser and Ferro found a mortality or long-term disability in around 15% of cases, often related to concomitant factors (e.g., cancer, infection, or thrombosis in other sites) [2]. Moreover, regarding the temporal relationship between CVT onset and thyrotoxicosis, 58.5% of the described cases had no history of thyroid disease, which was diagnosed during hospitalization, as in the patient we described. In contrast, 39.5% of the remaining patients had a history of thyroid dysfunction, but in all these cases, hyperthyroidism was either not well controlled with the ongoing therapy or the patients had spontaneously discontinued medications. This finding suggests that if thyroid disease is adequately controlled, the development of complications, including CVT, may be unlikely, and this is consistent with the work of Van Zaane et al. described below (i.e., the higher the doses of circulating thyroid hormone, the greater the alteration in coagulation factors) [63]. In contrast, only 2% of the described patients had hypothyroidism (i.e., Hashimoto’s disease). In hypothyroidism, both hypercoagulability and hypocoagulability have been reported, depending on the degree of thyroid dysfunction, severe hypothyroidism being more frequently associated with hypocoagulability [64,65,66]. It seems to be associated with several dysfunctions in the coagulation cascade and in platelet adhesion and functioning, in particular: reduced platelet count, adhesion, and aggregability; reduced values of vWF (i.e., acquired von Willebrand syndrome), FVIII, FIX, and factor XI (FXI), reduced fibrinogen, and increased fibrinolysis [67,68]. Moderate hypothyroidism, subclinical hypothyroidism, and autoimmune thyroid diseases, on the other hand, would appear to be more frequently associated with a prothrombotic state, due to increased fibrinogen, FVIII, and PAI-1 levels, increased mean platelet volume, decreased antithrombin-III, and a global decrease in fibrinolytic activity [64,69,70,71,72,73]. Moreover, hypothyroidism also seems to contribute to a slowing in venous flow with an indirect mechanism, leading to a left ventricular dysfunction and, consequently, to a decrease in venous velocity. Hypothyroidism-induced endothelial damage has already been reported for arterial vessels, as well as an increased prevalence of anti-endothelial cells antibodies in hypothyroid patients, suggesting a similar mechanism of damage on the venous side [74].

Regarding risk factors, 61% of the patients had other predisposing conditions for CVT in addition to hyperthyroidism. The most common risk factor was thrombophilia, which was found in 30% of patients. This could be further distinguished into genetic thrombophilia (e.g., deficiency of antithrombin III, protein C, and protein S, factor V Leiden positivity, or mutations in the methylenetetrahydrofolate reductase (MTHFR) gene) and acquired thrombophilia (e.g., antiphospholipid syndrome, hyperhomocysteinemia, or elevation of lupus anticoagulant). Specifically, out of all patients with coagulopathy, 23% had anticardiolipin antibodies, while equally distributed were the presence of lupus anticoagulant, MTHFR gene mutation, antithrombin III and protein C and S deficiency (15.4% each). Finally, hyperhomocysteinemia and Leiden factor were present in 7.7% of the cases each. Interestingly, two of the described patients had an history of malignancy, of which one patient had a papillary thyroid carcinoma in the context of hyperthyroidism, suggesting a dual pathogenetic mechanism of thrombosis [44]. A similar dual etiopathogenetic mechanism could be hypothesized in the case of a patient described by Mouton et al. who experienced CVT in the context of thyrotoxicosis related to postpartum thyroiditis [10]. Another factor favoring the development of thrombotic phenomena in women is oral contraceptive or hormone replacement therapy, as observed in a large proportion of female patients described in the analyzed case reports (33%). Other prothrombotic risk factors reported in the analyzed literature include smoking, sepsis, and head trauma. Interestingly, a case recently described by Gong et al. highlights how, even when other risk factors for CVT are present, the study of thyroid function and the possible finding of hyperthyroidism may be necessary to explain a patient’s clinical worsening. In this paper, the authors describe the case of a 29-year-old male who accessed the emergency department for a head injury following a fall from a height. The patient had developed a mild cerebral contusion with concomitant brain hemorrhage and was admitted to the neurosurgery department. Nine days after the event, he developed hyperthermia, hypertensive crisis, tachycardia, profuse sweating, and deterioration of consciousness to deep coma. The anamnestic finding of untreated Graves’ disease led clinicians to consider the hypothesis of a possible thyroid storm. Blood tests for thyroid function were consistent with this, and a CT-venography documented the presence of massive CVT. Antithyroid and anticoagulant therapy were started, and the patient gradually recovered [42]. This case represents a striking example of how alterations in thyroid function can contribute to the development of CVT even in conditions where other factors could be considered responsible. Given the link between the two conditions (i.e., altered thyroid function and CVT), the exclusion of thyrotoxicosis should be considered in all patients with suspected CVT, particularly if there is a history of hyperthyroidism, especially if poorly controlled or untreated; in the case of young women who are at greater risk of developing autoimmune thyroiditis; in patients with history of other concomitant autoimmune disorders; and in the case of conditions that may favor altered thyroid function (e.g., pregnancy). A thyroid function screening should also be performed in all CVT patients with a family history of thyroid disease, a high dietary iodine intake, an intake of thyrotoxic drugs (e.g., lithium, interferon α, or amiodarone) [75,76], who are smokers, have experienced recent psychological stress, or have neoplasms (i.e., ectopic thyroid hormone production) [77].

Interestingly, as in the case we described, 49% of patients did not appear to have any other predisposing factors for venous thrombosis, except for altered thyroid function. As early as the late 1990s, the first cases of hyperthyroidism with associated systemic hypercoagulability and increased risk of venous thrombosis were described. Several abnormalities of the coagulative and fibrinolytic systems have been described in this context, although a clear pathway underlying the mechanism has not yet been fully elucidated. One of the most widely described possible mechanisms for CVT, in the past, was mechanically based, related to the presence of goiter, which was thought to reduce venous outflow from the cranial district [65,78]. However, the subsequent description of CVT in the context of hyperthyroidism in patients who did not have goiter prompted a search for additional possible causal mechanisms. Specifically, a recent literature review examined the mechanism that might link the pro-coagulative state to altered thyroid function [79]. In particular, the importance of thyroid hormone levels on coagulative function has been demonstrated by the presence of significantly increased levels of vWF, fibrinogen, and D-dimer, even in patients with subclinical hyperthyroidism, when compared to euthyroid patients [80]. Moreover, in the context of hyperthyroidism, various alterations in platelet function, structure, and adhesion have been described [81,82,83]; in particular, increased platelet plug formation at baseline has been observed in patients with hyperthyroidism compared to healthy controls [84]. Thyroid hormones, and in particular T4, would also appear to activate the endothelium and thus promote platelet adhesion [85]. The increased platelet aggregability, as hypothesized by Homoncik et al., would appear to be related to an increase in plasma of vWF, linked to a probable genomic effect of T3 [84,85]. Concerning the action of thyroid hormones on coagulation factors, several studies have shown an increase in plasma levels of FVIII, with their subsequent normalization following antithyroid therapy [13,86]. Interestingly, in this regard, a study by Van Zaane et al. demonstrated how the induction of hyperthyroidism following levothyroxine administration in healthy volunteers caused an increase in FVIII, vWF, and PAI-1 levels [63]. Specifically, subjects were given different doses of levothyroxine, demonstrating that, in the case of lower doses, an increase in vWF alone was observed, while a higher dose also increased the activity of other coagulation factors, including FVIII [63]. This finding would seem to suggest an effect on coagulation dependent on blood concentrations of thyroid hormone (i.e., the higher the hormone concentration, the greater the risk of thrombosis). Coherently, an increase in FVIII activity was reported in 32% of the cases analyzed in this work. However, this finding was unfortunately not evaluated in our patient. Furthermore, another study in healthy volunteers consistently showed that an excess in thyroid hormones resulted in a hypofibrinolytic condition and an enhanced activated thrombin-activatable fibrinolysis inhibitor (TAFIa)-dependent prolongation of clot lysis [87]. Of note, recent studies have also shown an increase in factor XIIIB (FXIIIB), FIX (inhibitor of activated protein C), and alpha2-antiplasmin [11,12]. These elements are congruent with a prothrombotic and hypofibrinolytic condition. Indeed, the seminal systematic review and meta-analysis by Stuijver et al. confirmed that hyperthyroidism shifts the hemostatic balance towards hypercoagulability and hypofibrinolysis in both endogenous and exogenous hyperthyroidism [13]. Coherently, another study reported that subjects with hyperthyroidism exhibited increased clot maximum absorbance in comparison to a control group, along with prolonged clot lysis time, and this was found to be positively associated with FT4 levels [88]. Taken together, all these events also contribute to the formation of more resistant clots, given the increased fibrin network and reduced fibrinolysis [85]. Therefore, to summarize, thyroid hormones act at different levels in determining hypercoagulability: they cause an increase in FVIII, in particular in its B subunit (FVIIIB), FIX, vWF, fibrinogen and PAI-1, acting directly on hepatocytes and endothelial cells, causing a state of hypercoagulability and hypofibrinolysis [11,85,89,90,91]. T4 levels, which are closely correlated with increased FVIIIB, FIX (natural inhibitor of activated protein C), SERPIN A5, and alpha2-antiplasmin, and the negative correlation of plasminogen (the precursor of fibrinolysis), thus increase the risk of thrombosis [13,86]. T4 also appears to act on the cytokine cascade, promoting the release of pro-inflammatory cytokines, including interleukin-1 (IL-1), which increases the production of ultra large vWF multimers, IL-6, and IL-8. IL-1 gene expression is also regulated by thyroid hormone homologues, including tetraiodothyroacetic acid, which, in turn, regulates the transcription of the CX3CL1 chemokine gene [92,93]. The latter is released by endothelial cells in response to inflammatory factors and increases platelet adhesion [85]. Finally, T4 crosses the blood–brain barrier by binding with transthyretin (TTR) at the choroid plexuses, promoting the cerebral and cerebrospinal fluid uptake of the hormone [94,95]. This uptake plays an important role in brain development, but, when excessive, also has negative effects due to its pro-coagulative action [96]. Thus, the risk of thrombosis has been reported in series of patients with hyperthyroidism. Thyrotoxicosis therefore should be recognized by the clinician as a possible underlying cause of CVT, particularly when other prothrombotic factors are not found. In contrast to several other risk factors for CVT, most thyroid disorders are easily and effectively treatable, and the detection of any thyroid dysfunction underlying CVT could lead to the latter being considered as “provoked” which shortens the duration of anticoagulant therapy. Therefore, thyroid dysfunction should be considered as a relevant risk factor for CVT in clinical practice, and it would be useful to evaluate thyroid parameters in all patients with CVT. Another interesting point might be to consider thyroid dysfunction as a consequence of CVT. Few cases have been reported in the literature of cavernous sinus thrombosis, usually bilateral, determining hypopituitarism. Often, this is a septic thrombosis, with the extension of infection to the hypothalamic–pituitary site. The resulting disruption of the hypothalamic–pituitary gland axis has been associated with multiple hormonal dysfunctions, including thyroid dysfunction [97,98,99,100].

It is also important to point out that CVT in patients with autoimmune thyroid diseases might be justified by factors that can exert an action not only on the former condition, but also on the latter. Indeed, it has been shown that smoking can exert an influence on thyroid hormone levels and, in particular, it is an established risk factor for Graves’ disease according to a literature meta-analysis [101,102,103]. In addition, hyperthyroidism tends to predominate in females probably also because of hormonal factors, as shown by the presence of polymorphisms in the estrogen receptor that could have a pathophysiological link [104,105,106]. Indeed, during the menstrual cycle, pregnancy, and menopause, variations in estrogen levels could be correlated with the fluctuation of the disease. Furthermore, although the presence of a simple epiphenomenon cannot be excluded, Nabriski et al. found that anti-phospholipid antibodies are more frequent in patients with autoimmune thyroid disease than in healthy controls, leaving open the possibility that further studies could identify a causal link [107]. Taken together, this evidence suggests that some of the established risk factors for CVT may also contribute, at least in part, to the pathophysiology of thyroid disorders, opening up the intriguing possibility that CVT is not a direct consequence of altered thyroid hormone levels, but that both are a manifestation of an underlying risk condition. However, although suggestive, this hypothesis does not seem entirely justifiable, considering the discrepancy in population prevalence between the above-mentioned risk factors (extremely common), autoimmune thyroid disease (quite common, particularly in certain population groups), and CVT (relatively rare) [104,108,109,110]. It is therefore more reasonable to hypothesize a synergistic effect between risk factors common to both conditions and thyroid pathology in determining the onset of CVT, especially considering the cases (including those described in this review) in which thyroid pathology is the sole risk factor for CVT, in the absence of further evidence to justify the two conditions. 

Of note, CVT is not only linked to numerous autoimmune diseases, but also to vaccination against various pathogens [111,112]. This pathophysiological relationship has gained great prominence recently, as repeated cases of vaccine-induced thrombocytopenia and thrombosis (VITT) have been reported in connection with adenovirus-based COVID-19 vaccines that have been used to cope with the pandemic emergency that has affected the world in recent years [113,114,115,116,117]. Although some of the patients initially reported in the literature had the common characteristics associated with CVT (e.g., females, obesity, use of contraceptive therapies or hormone replacement therapies), as argued by a recent review on the subject, the pathogenesis appears to be different from classic CVT, as many patients did not have pre-existing prothrombotic risk factors at the time of vaccination [118,119,120]. Moreover, importantly, the prognosis of these patients was extremely severe, since mortality exceeding 50% and, in general, poor functional recovery were observed [119]. The pathogenesis of this condition is still largely unknown, although various possible mechanisms have been speculated, among which the presence of high values of IgG antibodies against platelet factor 4 antibodies (PF4)–polyanion complexes, which seem to be present in all CVT-VITT patients and which could be, according to some authors, the main driver behind the severity of the disease, regardless of the presence of other prothrombotic risk factors, seems to be the most relevant [118]. However, other authors have hypothesized a mechanism related to the possibility that COVID-19 adenoviral vaccines might induce the production of both natural and optimized versions of the SARS-CoV-2 spike protein, leading to different splicing processes which could interact with cells expressing ACE2 receptors, including platelets and endothelial cells, potentially causing their activation and, ultimately, promoting the creation of a prothrombotic environment [121]. Furthermore, other authors speculate about a direct binding of the adenoviral vector to platelet receptors, responsible for their activation and aggregation [122]. However, irrespective of the pathogenetic mechanism, these data make it possible to hypothesize that CVT may not only develop without a pre-existing prothrombotic background, as in the case we have reported and in other literature evidence, but may even be associated with worse outcomes than CVT cases in the context of conventional risk factors. This seems to indicate an important difference between CVT-VITT cases and those related to autoimmune disease (such as the one presented in this report), which are generally associated with a better outcome than the former case. For example, a description of 64 patients with CVT in the context of neuro-Behçet’s disease, i.e., vasculitis of unknown etiology characterized by mucocutaneous, ocular, arthritic, and vascular manifestations, reported death in only 6.2% of the patients examined, although the number of long-term sequelae is as high as 35.9% [123]. A similar argument can also be applied to six cases of CVT reported in the context of systemic lupus erythematosus (SLE), five of which recovered rapidly with steroids and heparin, and the same conclusion was reached for a recent review of literature cases of CVT associated with Sjögren’s syndrome, a chronic inflammatory autoimmune disease characterized by lymphocyte infiltration of the exocrine glands leading to xerophthalmia and xerostomia, all of which had favorable outcomes [124,125]. The reason why the prognosis of these patients is better than CVT-VITT is still unclear, as further studies are needed to understand the pathophysiology of VITT and explain the associated very high mortality risk. However, it is possible that the mechanism associated with adenoviral vaccine for SARS-CoV-2 is more violent and abrupt than that present in autoimmune diseases, which often require a trigger (e.g., hormonal therapies, obesity, pregnancy, or smoking) to unleash their thrombogenic potential [118,126]. 

To summarize, in the diagnostic work-up of CVT, as for other conditions with multifactorial and complex etiopathogenesis, it is advisable in clinical practice to use a strategic and systematic approach, first searching for more probable causes of disease, later broadening the spectrum, looking for rarer but still plausible conditions. An appropriate diagnostic procedure should therefore include a search for genetic or acquired coagulopathies, autoimmune conditions, history of vaccination, possible neoplasms, systemic or local infections, history of trauma, use of oral contraceptives or hormone replacement therapy, and conditions such as pregnancy and puerperium in women. In this context, given the growing evidence, screening for thyroid function should also be performed, particularly in patients with a history of other autoimmune diseases, female, of a young age, and also in individuals with no history of thyroid disease. Such an approach should help the clinician identify all possible underlying causes of a CVT in order to target the therapy and avoid any recurrence.

As a partial limitation for our study, we have chosen to only present case reports and case series because, given the rarity and peculiarity of the condition examined, a level of individual characterization was preferable [127]. However, this certainly prevented us from including cohort studies such as that of Ferro et al., which enrolled an international cohort of 624 CVT patients, reporting a thyroid dysfunction in 11 of them, which appears consistent with the rarity of the condition, although, unfortunately, no details are available on the thyroid disorder nor on the characteristics of the patients in this category [128]. Another limitation is that some case descriptions omitted crucial data to better understand the pathophysiology and prognosis of the patients presented (e.g., data on FVIII activity, outcome, or concomitant coagulopathies). Furthermore, information on the risk of CVT recurrence in these patients is not available. Finally, many of the reported cases are dated, and there are not many recent reports available on CVT in the context of thyrotoxicosis.

Further studies could be useful to assess the possible risk of recurrence in patients with previous CVT and hyperthyroidism, particularly in the case of poorly controlled thyroid function, and it would be useful to update the literature in order to collect, given the relative rarity of the condition, all available evidence on the subject.

## 4. Conclusions

The available evidence in the literature regarding the association between CVT and thyrotoxicosis involves a few case reports and case series which are not quite recent. This paper has the advantage of collecting the different available pieces of evidence, comparing them with each other and with the case described, and discussing the possible implications regarding considering thyroid dysfunction as a possible risk factor for CVT, in order to gain resonance in clinical practice.

Based on the currently available evidence, thyroid hormones would appear to affect coagulation and fibrinolysis, inducing, in the case of hyperthyroidism, a prothrombotic and hypofibrinolytic state through the activation of several coagulation factors, the release of proinflammatory cytokines, and the increase in platelet aggregability and adhesion. Consequently, although CVT in most cases presents a multifactorial etiology and there are more common risk factors underlying its development, altered thyroid function should be considered as a possible causative agent, particularly in patients with no other risk factors. The evaluation of thyroid function could therefore be introduced as a screening tool in the etiopathological framing of patients with CVT. At the same time, patients with diagnosed thyrotoxicosis showing severe headache, acute symptomatic seizures, increased intracranial pressure, or altered consciousness should be investigated for the presence of CVT.

## Figures and Tables

**Figure 1 jpm-13-01557-f001:**
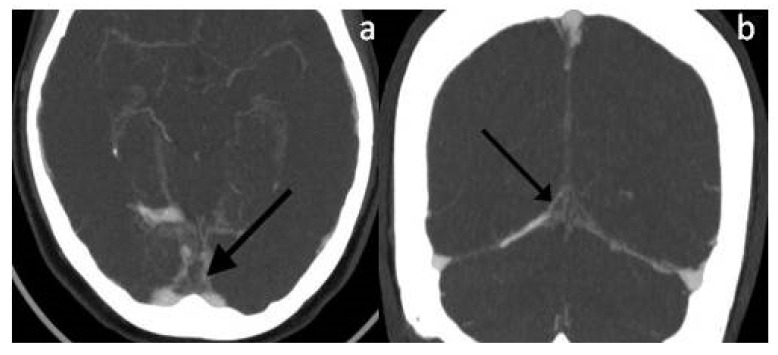
(**a**,**b**) Maximum intensity projection (MIP) post-contrast CT-venography reconstruction showing the ‘empty delta’ sign, i.e., occlusion of the confluence of sinuses (see arrows).

**Figure 2 jpm-13-01557-f002:**
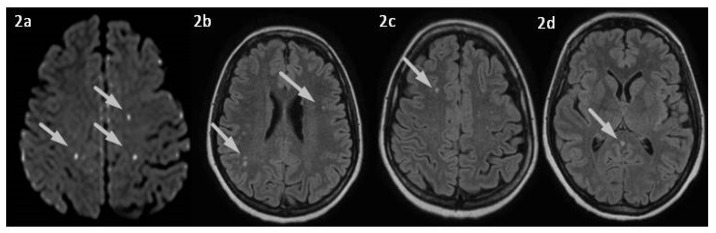
(**a**–**d**) Brain MRI. Diffusion-weighted imaging (DWI) (**a**), Fluid Attenuated Inversion Recovery (FLAIR) sequences showing (**b**,**c**) small multifocal acute ischemic lesions, and (**d**) spontaneous hypersignal of Vein of Galen suggesting thrombosis (arrow).

**Figure 3 jpm-13-01557-f003:**
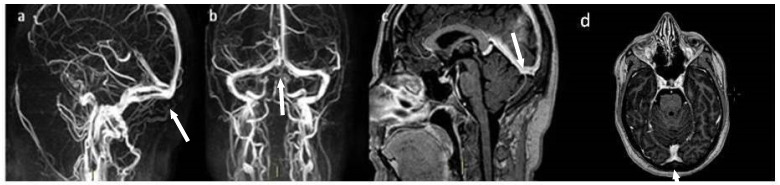
(**a**–**d**) Brain MRI. (**a**,**b**) Venous 3D MIP phase contrast (PC), (**c**,**d**) 3D contrast-enhanced T1-weighted images showing a complete recanalization of the previously involved venous sinuses (see arrows).

**Figure 4 jpm-13-01557-f004:**
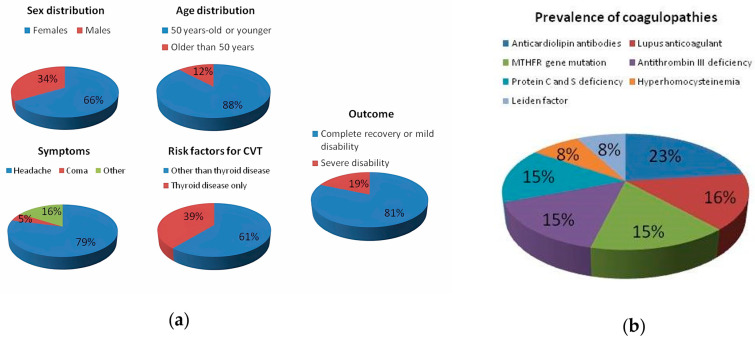
(**a**) The graphs show the main demographic and clinical characteristics of the described patients. (**b**) The graph shows the prevalence of coagulopathies in the described patients.

**Table 1 jpm-13-01557-t001:** Summary of the main characteristics of the patients described in the included papers. Coagulopathy: genetic and/or acquired thrombophilia (e.g., deficiency of antithrombin III and protein C, S, Factor V Leiden, prothrombin gene mutation, hyperhomocysteinemia, antiphospholipid antibodies, or lupus anticoagulant). Abbreviations: Ap: aphasia; CI: cognitive impairment; CR: complete recovery; CV: cortical veins; DCV: deep cerebral veins; F: female; FS: focal seizures; GS: generalized seizures; H: headache; ISS: inferior sagittal sinus; LHe: left hemiparesis; LIJV: left internal jugular vein; LMWH: low molecular weight heparin; LSS: left sigmoid sinus; LTS: left transverse sinus; LVFD: left visual field defect; M: male; MD: mild disability; NA: not assessed; PI: progressive improvement; RHe: right hemiparesis; RIJV: right internal jugular vein; RSS: right sigmoid sinus; RTS: right transverse sinus; RVFD: right visual field defect; S: seizures; SD: severe disability; SS: straight sinus; SSS: superior sagittal sinus; UH: unfractionated heparin; V: vomiting.

First Author/Year	Age/Sex	Neurological Symptoms	Site of Thrombosis	FVIII Activity	Acute Treatment	Oral Contraceptives	Smoker	Coagulopathy	Malignancies	Infections	Graves’ Disease	Other Prothrombotic Conditions	Outcome	References
Yokoyama, 2019	48 F	Fever, H	SSS	Increased	LMWH	-	-	-	-	Viral meningitis	-	-	CR	[22]
Kraut, 2017	62 F	H, GS	SSS, RTS	Increased	LMWH	-	-	-	-	-	+	-	CR	[23]
Hieber, 2016	52 F	H	LSS, LTS	-	LMWH	-	-	+	-	-	+	-	MD	[24]
Srikant, 2013	42 F	H, drowsiness, LHe	RSS, RTS	NA	LMWH + decompressive craniectomy	NA	Na	+	NA	NA	+	-	NA	[25]
Anuszkiewicz, 2021	15 M	H, RHe, Ap	SSS, LTS, LSS	Increased	UH	/	-	-	-	-	+	-	CR	[26]
Mouton, 2005	32 F	H, vertigo, right arm paresthesia	LTS	Increased	UH	+	-	-	-	-	- (post-partum thyroiditis)	Puerperium	CR	[10]
Mouton, 2005	49 F	H, left arm weakness, dysarthria	SSS, RTS	Increased	NA	+	-	-	-	-	+	-	CR	[10]
Mouton, 2005	50 F	H, blurred vision, RVFD	CV, LTS	Increased	NA	-	+	-	-	-	+	-	CR	[10]
Mouton, 2005	39 F	H, FS	SSS, RTS	Increased	NA	-	+	-	-	-	+	-	CR	[10]
Hermans, 2011	22 F	GS	LSS	-	UH	+	-	-	-	-	+	-	PI	[27]
Elhassa, 2020	41 M	GS	SSS, CV	-	LMWH	/	-	-	-	-	+	-	CR	[28]
Tanabe, 2017	49 F	H, LHe	LTS, LSS, LIJV	-	UH	-	-	-	-	-	+	-	PI	[29]
Waheed, 2016	48 F	Drowsiness, H, V	CV, SS	Increased	UH	-	-	+	-	-	+	-	NA	[9]
Rehman, 2018	31 M	Drowsiness, H, V	SSS, SS, RTS, LTS	-	LMWH	/	-	+	-	-	+	-	CR	[30]
Chee, 2020	40 F	H, Ap	LSS, LTS, LIJV	NA	LMWH+ decompressive craniectomy	-	-	-	+	-	+	-	NA	[31]
Knudsen-Baas, 2014	17 F	H, S, coma	SSS, RTS, RSS, RIJV	NA	UH	+	-	+	-	-	+	-	CR	[32]
Gomes, 2021	23 F	H, V, GS	RTS, RSS, SS	NA	LMWH	-	-	-	-	-	+	-	CR	[33]
Kim, 2013	23 F	NS	NS	NA	NS	+	-	-	-	-	+	-	NA	[34]
Pekdemir, 2008	28 M	H, V, papilledema	LSS, LTS	NA	UH	/	NA	NA	NA	NA	- (chronic thyroiditis)	NA	MD	[35]
Bensalah, 2011	23 M	H	SSS, RTS, RSS	NA	LMWH	/	NS	-	-	-	+	Steroid therapy	NA	[36]
Hwang, 2012	31 M	H, S, CI	SSS	-	Warfarin	/	-	-	-	-	-	-	CR	[37]
Ra, 2001	60 M	H, GS, LHe	SSS, LTS	-	Urokinase	/	NA	-	-	-	+	-	PI	[38]
Kim,2016	39 M	GS, LHe	SSS	-	UH	/	-	+	-	-	+	-	CR	[39]
Liu, 2015	44 F	H, cortical blindness	LTS, LSS	Increased	UH	+	-	+	-	-	+	-	SD	[40]
Fandler-Hofler, 2022	60 F	H	LTS, LSS	-	LMWH	-	-	-	-	-	+	-	CR	[41]
Fandler-Hofler,2022	33 F	H	LTS, SS	-	LMWH	+	-	-	-	-	+	-	CR	[41]
Gong, 2022	29 M	Coma	Multiple sites		LMWH	/	-	+	-	-	+	Head trauma	MD	[42]
Jia, 2022	44 F	H, drowsiness, RHe	RTS, RSS, SS, SSS, LTS, LSS	-	LMWH, urokinase, alteplase, thrombus aspiration	-	-	-	-	-	+	-	PI	[43]
Migeot, 2013	26 F	H, LHe, LVFD	SSS, RTS	Increased	Warfarin	-	-	-	+ (papillary thyroid carcinoma)	-	+	Puerperium	CR	[44]
Verberne, 2000	28 F	Drowsiness	LTS, SS, LIJV	Increased	LMWH	+	-	+	-	-	+	-	CR	[45]
Son, 2019	31 M	S	SSS, RTS, RSS	NA	LMWH	/	NA	-	-	-	+	-	CR	[46]
Aggarwal, 2013	44 F	H, V, RHe	SSS, SS	-	LMWH	-	-	-	-	-	- (Hashimoto thyroiditis)	-	CR	[47]
Janovsky, 2013	21 F	Ap, RHe	SSS, LTS, LSS	Increased	LMWH	-	-	+	-	-	+	-	MD	[48]
Dai, 2000	39 M	H, GS	SSS	-	LMWH	/	NA	-	-	-	NA	-	CR	[49]
Elbers, 2014	50 F	Ap, RVFD	LSS, SS	-	LMWH	-	NA	-	-	-	+	-	MD	[50]
Maes, 2002	39 F	S, H, confusion	LTS, LIJV	Increased	UH	+	-	+	-	-	+	-	CR	[51]
Madan, 2018	28 F	H, right vision loss	LSS, LTS	NA	LMWH	-	NA	-	-	-	+	-	CR	[52]
Silburn, 1996	18 F	H, confusion, fever, neglect	DCV, ISS	NA	NS	+	-	-	-	-	+	-	NA	[53]
Situmeang, 2022	37 M	H, fever	SSS, RTS, RSS	NA	LMWH	/	-	-	-	COVID-19	+	-	CR	[54]
Strada,2008	29 M	H, S, LHe	SSS, RTS	NA	UH	/	-	+	-	-	+	Hyperhomocisteinemia	CR	[55]
Tashiro, 2023	38 F	H, bilateral VI palsy, papilledema, diplopia	SSS, RTS, RSS	Increased	UH	-	-	-	-	-	+	-	CR	[56]
Usami, 2009	34 F	H, V, diplopia, LHe	SSS, RTS, SS, DCV	Increased	UH, plasma exchanges	-	-	+	-	-	+	-	SD	[57]
Van Eimeren, 2012	8 F	V, H	Massive CVT	Increased	LMWH	/	/	+	-	-	+	Dehydration	MD	[58]

## Data Availability

Not applicable.

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
