# Peer review of "Cerebral Venous Thrombosis during Thyrotoxicosis: Case Report and Literature Update"

_jpm, 2023, doi:10.3390/jpm13111557_

Round 1
Reviewer 1 Report
Comments and Suggestions for Authors
Thank you for the opportunity to review this interesting case report and literature update about CVT in the context thyrotoxicosis. It is a case of a patient with concomitant thyrotoxicosis and an overview of 39 other published cases. It is a meticulous report, with some added value to the field. Conceptually it is an interesting manuscript. Execution requires a few minor changes.
Thyroid disorders are more often described as associated with CVT. It is interesting to show whether they are different to other types of CVT. As we know another autoimmune type of CVT is, CVT-VITT is largely different to a classical CVT. Could this also be the case of other autoimmune diseases associated-CVT? Are these cases more serious - or more benign? which population is most at risk?
The major challenges is also to show that 1) given high prevalence of thyroid disorders in the population at risk (middle-aged women) it is not an expected background rate what is observed 2) there is a plausible pathophysiological association. As you explain in discussion, there are possible pathophysiological mechanisms (induced procoagulative state). On the other hand, could it be that thyroid problems are a manifestation of an underlying problem/condition which causes CVT?
Improvement points/questions
Concept: is there a reason you did not include case series/cohort studies and only case reports? Did you consider adding cases from cohort studies (ISCVT as an example, Ferro Stroke 2004)?
Case description:
Imaging section: Did you make a non-contrast head CT? What was seen there?
Laboratory results: was the patient screened for infection?
When was the dental extraction surgery performed, could it be related?
Treatment: why was warfarin started? What was the reasoning behind this choice?
Figures: please annotate your images in a more detailed manner. Please state which scan and sequence you are using, what exactly you see and where do you see the changes. E.g. figure 1. Perhaps CT-venography would be more appropriate? Also consider adding an arrow to the second image in figure 1. Perhaps name them 1A and 1B? (If adding 1B is in your opinion of added value?) MRI: Consider adding arrows. Can you say something about diffusion restriction? Do you have more interesting sequences to show? Also: I am not sure I would call this “acute phase” given that a patient had symptoms for weeks - perhaps subacute would be more appropriate?
Tables:
Smoke as a name of a column is insufficient: either smoker or smoking.
What exactly is meant by coagulopathy? Consider to specify in footnote.
Discussion
- Discussion can be shortened and more focused on the interesting discussion points. E.g. no need to elaborate on the fact that contraception/pregnancy/postpartum are known risk factors.
- Consider using specific proportions based on the data you gathered instead of general terms like “a few cases” or “ in most cases”.
- It would be interesting to know whether these cases are in any way different to non-thyroid/non-autoimmune related CVT?
- Adding an interesting case description in your discussion should form a clear purpose for a reflection/conclusion/question. For instance case with head injury, deterioration and hyperthyroidism - suggests that indeed many patients have more than one risk factor and it is important to screen for thyroid disorders (particularly after neurological worsening) even when there already seems to be one risk factor.
- Could you say something about onset of thyrotoxicosis - is it acute, parallel to CVT, or can chronic thyroid problems induce CVT?
- Please mention which patients in your opinion are most at risk and should be particularly screened for thyroid disorders. are there specific comorbidities?
- Please comment on the outcomes.
- Consider adding limitations and strength section and to elaborate a bit more on which directions should the future studies focus on. How to study this optimally?
Conclusions section is appropriate. Thyroid abnormalities in CVT patients seem to be part of multifactorial etiology of this disease. No further comments.
Other minor points:
Line 39 “Brain stroke” seems a bit redundant. Stroke is sufficient
Line 147: Unclear what you mean by “ thrombosis related neurological symptoms are associated with thyrotoxicosis symptoms”, as I see it some of these manifestations could be related to either one or both.
160: I believe unfractionated heparin?
165-168: Were all these coagulopathies present in your patients? More interesting would be to mention which coagulopathies were in fact most common in your study.
Comments on the Quality of English Language
Please see section above. No further comments.
Author Response
Thyroid disorders are more often described as associated with CVT. It is interesting to show whether they are different to other types of CVT. As we know another autoimmune type of CVT is, CVT-VITT is largely different to a classical CVT. Could this also be the case of other autoimmune diseases associated-CVT? Are these cases more serious - or more benign? which population is most at risk?
We thank the reviewer for this insightful suggestion, which allowed us to further explore the link between CVT and auto-immune disorders. Accordingly, we added this in the Discussion section:
“Of note, CVT is not only linked to numerous autoimmune diseases, but also to vaccination against various pathogens [102]. This pathophysiological relationship has gained great prominence recently, as repeated cases of vaccine-induced thrombocytopenia and thrombosis (VITT) have been reported in connection with adenovirus-based Covid-19 vaccines that have been used to cope with the pandemic emergency that has affected the world in the last years [103,104]. Although some of the patients initially reported in the literature had the common conditions associated with CVT (e.g. females, obesity, use of contraceptive therapies or hormone replacement therapies), as argued by a recent review on the subject, the pathogenesis appears to be different from classic CVT, as many patients do not have pre-existing pro-thrombotic risk factors at the time of vaccination [105-107]. Moreover, importantly, the prognosis of these patients was extremely severe, since mortality exceeding 50% and, in general, poor functional recovery were observed [106]. The pathogenesis of this condition is still largely unknown, although various possible mechanisms have been speculated, among which the presence of high values of IgG antibodies against platelet factor 4 antibodies (PF4)-polyanion complexes, which seem to be present in all CVT-VITT patients and which could be, according to some authors, the main driver of the severity of the disease regardless of the presence of other pro-thrombotic risk factors, seems to be the most relevant [105]. However, other authors have hypothesised a mechanism related to the possibility that Covid-19 adenoviral vaccines might induce the production of both natural and optimised versions of the SARS-CoV-2 spike protein, leading to different splicing processes which could interact with cells expressing ACE2 receptors, including platelets and endothelial cells, potentially causing their activation and, ultimately, promoting the creation of a pro-thrombotic environment [108]. Furthermore, other authors speculate about a direct binding of the adenoviral vector to platelet receptors, responsible for their activation and aggregation [109]. However, irrespective of the pathogenetic mechanism, these data make it possible to hypothesise that CVT may not only develop without a pre-existing pro-thrombotic background, as in the case we have reported and in other literature evidence, but may even be associated with worse outcomes than CVT cases in the context of conventional risk factors. This seems to indicate an important difference between CVT-VITT cases and those related to autoimmune disease (such as the one presented in this report), which are generally associated with a better outcome than the former case. For example, a description of 64 patients with CVT in the context of Neuro-Behçet's, i.e. vasculitis of unknown etiology characterised by mucocutaneous, ocular, arthritic, and vascular manifestations, reported death in only 6.2% of the patients examined, although the number of long-term sequelae is as high as 35.9% [110]. A similar argument can also be applied to 6 cases of CVT reported in the context of systemic lupus erythematosus (SLE), 5 of which recovered rapidly with steroids and heparin, and the same conclusion was reached for a recent review of literature cases of CVT associated with Sjögren's syndrome, a chronic inflammatory autoimmune disease characterised by lymphocyte infiltration of the exocrine glands leading to xerophthalmia and xerostomia all of which had favourable outcomes [111,112]. The reason why the prognosis of these patients is better than CVT-VITT is still unclear, as further studies are needed to understand the pathophysiology of VITT and explain the associated very high mortality risk. However, it is possible that the mechanism associated with adenoviral vaccine for Sars-Cov2 is more violent and abrupt than that present in auto-immune diseases, which often require a trigger (e.g. hormonal therapies, obesity, pregnancy, smoking) to unleash their thrombogenic potential [105,113].”
The major challenges is also to show that 1) given high prevalence of thyroid disorders in the population at risk (middle-aged women) it is not an expected background rate what is observed 2) there is a plausible pathophysiological association. As you explain in discussion, there are possible pathophysiological mechanisms (induced procoagulative state). On the other hand, could it be that thyroid problems are a manifestation of an underlying problem/condition which causes CVT?
We thank the reviewer for this profound observation. In this regard, in addition to further elaborating on the pathophysiological link between auto-immune diseases and CVT, we have added a paragraph in the Discussion. In particular, please see:
“It is also important to point out that CVT in patients with autoimmune thyroid diseases might be justified by factors that can exert an action not only on the former condition, but also on the latter. Indeed, it has been shown that smoking can exert an influence on thyroid hormone levels and, in particular, it is an established risk factor for Graves' disease according to a literature meta-analysis [95-97]. In addition, hyperthyroidism tends to predominate in females probably also because of hormonal factors, as shown by the presence of polymorphisms in the estrogen receptor that could have a pathophysiological link [98,99]. Indeed, during the menstrual cycle, pregnancy, and menopause, variations in estrogen levels could be correlated with the fluctuation of the disease. Furthermore, although the presence of a simple epiphenomenon cannot be excluded, Nabriski et al. found that anti-phospholipid antibodies are more frequent in patients with autoimmune thyroid disease than in healthy controls, leaving open the possibility that further studies could identify a causal link [100]. Taken together, this evidence suggests that some of the established risk factors for CVT may also contribute, at least in part, to the pathophysiology of thyroid disorders, opening up the intriguing possibility that CVT is not a direct consequence of altered thyroid hormone levels, but that both are a manifestation of an underlying risk condition. However, although suggestive, this hypothesis does not seem entirely justifiable, considering the discrepancy in population prevalence between the above-mentioned risk factors (extremely common), auto-immune thyroid disease (quite common, particularly in certain population groups) and CVT (relatively rare) [98,101]. It is therefore more reasonable to hypothesise a synergistic effect between risk factors common to both conditions and thyroid pathology in determining the onset of CVT, especially considering the cases (including those described in this review) in which thyroid pathology is the sole risk factor for CVT, in the absence of further evidence to justify the two conditions”.
Improvement points/questions
Concept: is there a reason you did not include case series/cohort studies and only case reports? Did you consider adding cases from cohort studies (ISCVT as an example, Ferro Stroke 2004)?
We thank the reviewer for the suggestion. The choice not to include larger cohort studies stems from the fact that, in many cases, especially when dealing with large cohorts, it is more difficult to obtain a high level of individual patient characterisation. Since this is a relatively rare condition, we preferred an approach that would allow us to detail each individual case presented in depth. However, considering the value added by this information, we have added the following in the Limits section:
“As a partial limitation for our study, we have chosen to only present case reports and case series because, given the rarity and peculiarity of the condition examined, a level of individual characterisation was preferable [114]. However, this certainly prevented us from including cohort studies such as that of Ferro et al., which enrolled an international cohort of 624 CVT patients, reporting a thyroid dysfunction in 11 of them, which appears consistent with the rarity of the condition, although, unfortunately, no details are available on the thyroid disorder, nor on the characteristics of the patients in this category [115].”
Case description:
We thank the reviewer for the suggestion, we clarified these aspects in Case presentation. In particular:
Imaging section: Did you make a non-contrast head CT? What was seen there?
“and brain computerized tomography (CT), which was unremarkable”.
Laboratory results: was the patient screened for infection?
“Laboratory tests for any infectious aetiology were negative too”.
When was the dental extraction surgery performed, could it be related?
“performed several years before”. This datum makes a causal link extremely unlikely.
Treatment: why was warfarin started? What was the reasoning behind this choice?
We chose to use warfarin due to the greater experience of our centre with this drug in treating CVT. DOACs could also be a reasonable option, but, being the diagnostic work up still in progress, the presence of concomitant factors (e.g. neoplasms) was not yet excluded, making this choice less safe, considering the scarcer evidence in the literature. Moreover, as far as we know, there are no large RCTs on the topic.
We added a sentence on this, please see Case presentation:
“Admitted to the Neurology Unit, since the diagnostic work-up was still in progress and considering direct anticoagulants (DOACs) less explored [17], she was started on anticoagulant with low-molecular-weight heparin (LMWH) at a body weight adjusted dose as a bridging-therapy, later replaced with warfarin.”
Figures: please annotate your images in a more detailed manner. Please state which scan and sequence you are using, what exactly you see and where do you see the changes. E.g. figure 1. Perhaps CT-venography would be more appropriate? Also consider adding an arrow to the second image in figure 1. Perhaps name them 1A and 1B? (If adding 1B is in your opinion of added value?) MRI: Consider adding arrows. Can you say something about diffusion restriction? Do you have more interesting sequences to show? Also: I am not sure I would call this “acute phase” given that a patient had symptoms for weeks - perhaps subacute would be more appropriate?
Thank you for your suggestion. We added more sequences and we modified the descriptions in a more detailed manner. Please see the Case presentation section.
Tables:
Smoke as a name of a column is insufficient: either smoker or smoking.
Thank you for the suggestion, we modified.
What exactly is meant by coagulopathy? Consider to specify in footnote.
We specified in the Table footnote as follows:
“Coagulopathy: genetic and/or acquired thrombophilia (e.g. deficiency of antithrombin III and protein C, S, Factor V Leiden, prothrombin gene mutation, hyperhomocysteinemia, antiphospholipid antibodies, lupus anticoagulant).”
Discussion
- Discussion can be shortened and more focused on the interesting discussion points. E.g. no need to elaborate on the fact that contraception/pregnancy/postpartum are known risk factors.
We agree, thus we removed some sentences already reported in the Introduction or redundant.
Consider using specific proportions based on the data you gathered instead of general terms like “a few cases” or “ in most cases”.
We thank the Reviewer for this useful suggestion. We added specific proportions in the Discussion, please see the Discussion section in the manuscript.
It would be interesting to know whether these cases are in any way different to non-thyroid/non-autoimmune related CVT?
We thank the Reviewer for this insightful suggestion. Please note that we addressed the difference between thyroid induced CVT and other autoimmune disorders in the previous comment. About non-autoimmune disorders, we added a sentence in the Discussion:
“In 81.4% of cases the outcome was favourable for all patients with complete recovery or only minimal disability. This proportion seems in line with those reported in the literature, regardless of CVT aetiology. Indeed, evaluating various meta-analyses, Bousser and Ferro found a mortality or long-term disability of around 15% of cases, often related to concomitant factors (e.g. cancer, infection, thrombosis in other sites) [2].”
Adding an interesting case description in your discussion should form a clear purpose for a reflection/conclusion/question. For instance case with head injury, deterioration and hyperthyroidism - suggests that indeed many patients have more than one risk factor and it is important to screen for thyroid disorders (particularly after neurological worsening) even when there already seems to be one risk factor.
Dear Reviewer, thank you for this suggestion. We have described in a more detailed manner the case of the young man with head injury and we added a more in-depth discussion regarding the association of hyperthyroidism and other risk factors in the development of CVT. Please see in the Discussion section:
“Interestingly, a case recently described by Gong et al. highlights how, even when other risk factors for CVT are present, the study of thyroid function and the possible finding of hyperthyroidism may be necessary to explain a patient's clinical worsening. In this paper, the authors describe the case of a 29-year-old male who accesses the emergency department for head injury following a fall from height. The patient had developed a mild cerebral contusion with concomitant brain hemorrhage and was admitted to the neurosurgery department. Nine days after the event, he developed hyperthermia, hypertensive crisis, tachycardia, profuse sweating, and deterioration of consciousness to deep coma. The anamnestic finding of untreated Graves' disease led clinicians to consider the hypothesis of a possible thyroid storm. Blood tests for thyroid function were consistent with this, and a CT-venography documented the presence of massive CVT. Antithyroid and anticoagulant therapy were started and the patient gradually recovered [40]. This case represents a striking example of how alterations in thyroid function can contribute to the development of CVT even in conditions where other factors could be considered responsible.”
Could you say something about onset of thyrotoxicosis - is it acute, parallel to CVT, or can chronic thyroid problems induce CVT?
Thank you for your valuable suggestion. We added a more detailed description on this point in the Discussion section, please see:
“Moreover, regarding the temporal relationship between CVT onset and thyrotoxicosis, 58.5% of the described cases had no history of thyroid disease, which was diagnosed during hospitalization, as in the patient we described. In contrast, 39.5% of the remaining patients had a history of thyroid dysfunction, but in all these cases hyperthyroidism was either not well controlled with the ongoing therapy or the patients had spontaneously discontinued medications. This finding suggests that if thyroid disease is adequately controlled, the development of complications, including CVT, may be unlikely and it is consistent with the work by Van Zaane et al. as described below (i.e. higher the doses of circulating thyroid hormone greater the alteration in coagulation factors) [61]. In contrast, only 2% of the described patients had hypothyroidism (i.e. Hashimoto's disease).”
Please mention which patients in your opinion are most at risk and should be particularly screened for thyroid disorders. Are there specific comorbidities?
Thank you for your insightful suggestion. We added some sentences in the Discussion section, please see:
“Given the link between the two conditions (i.e. altered thyroid function and CVT), the exclusion of thyrotoxicosis should be considered in all patients with suspected CVT, particularly if there is a history of hyperthyroidism, especially if poorly controlled or untreated, in the case of young women who are at greater risk of developing autoimmune thyroiditis, in patients with history of other concomitant autoimmune disorders, and in case of conditions that may favour altered thyroid function (e.g. pregnancy). A thyroid function screening should also be performed in all CVT patients with a family history of thyroid disease, a high dietary iodine intake, the intake of thyrotoxic drugs (e.g. lithium, interferon α, amiodarone) [72,73], who are smokers, have experienced recent psychological stress, or have neoplasms (i.e. ectopic thyroid hormone production) [74].”
Please comment on the outcomes.
Dear Reviewer, thank you for this suggestion. We have addressed this in previous comments, in particular emphasising that the outcome of CVT in the context of thyroid changes is generally favourable. Furthermore, we have compared, in terms of outcome, CVT cases regardless of aetiology and those due to different autoimmune diseases. Finally, we examined CVT-VITT and argued about its often unfavourable outcome and related mechanisms.
Consider adding limitations and strength section and to elaborate a bit more on which directions should the future studies focus on. How to study this optimally?
Dear Reviewer, we added a Limitations and Strengths section before the Conclusions. Please see:
“As a partial limitation for our study, we have chosen to only present case reports and case series because, given the rarity and peculiarity of the condition examined, a level of individual characterisation was preferable [114]. However, this certainly prevented us from including cohort studies such as that of Ferro et al., which enrolled an international cohort of 624 CVT patients, reporting a thyroid dysfunction in 11 of them, which appears consistent with the rarity of the condition, although, unfortunately, no details are available on the thyroid disorder, nor on the characteristics of the patients in this category [115]. Another limitation is that some case descriptions omitted crucial data to better understand the pathophysiology and prognosis of the patients presented (e.g. data on FVIII activity, outcome, concomitant coagulopathies). Furthermore, information on the risk of CVT recurrence in these patients is not available. Finally, many of the reported cases are dated and there are not many recent reports available on CVT in the context of thyrotoxicosis.
Further studies could be useful to assess the possible risk of recurrence in patients with previous CVT and hyperthyroidism, particularly in case of poorly controlled thyroid function, and it would be useful to update the literature in order to collect, given the relative rarity of the condition, all available evidence on the subject.”
Other minor points:
Line 39 “Brain stroke” seems a bit redundant. Stroke is sufficient
We corrected.
Line 147: Unclear what you mean by “ thrombosis related neurological symptoms are associated with thyrotoxicosis symptoms”, as I see it some of these manifestations could be related to either one or both.
We thank the Reviewer for this comment. We clarified:
“In most cases, patients presented a range of symptoms that could be related to CVT and/or thyrotoxicosis, such as tachycardia, heat intolerance, weight loss, nausea and vomiting, and diarrhea and most patients had no previous diagnosis of thyroid disorder.”
160: I believe unfractionated heparin?
Thank you, we corrected.
165-168: Were all these coagulopathies present in your patients? More interesting would be to mention which coagulopathies were in fact most common in your study.
Thank you, we added a sentence about this point in the Discussion section:
“Specifically, out of the total of patients with coagulopathy, 23% had anticardiolipin antibodies, equally distributed were the presence of lupus anticoagulant, MTHFR gene mutation, antithrombin III and protein C and S deficiency (15.4% each), and finally hyperhomocysteinemia and Leiden factor were present in 7.7% of the cases each.”
Finally, we revised the language with the help of a native speaker expert.
Reviewer 2 Report
Comments and Suggestions for Authors
I have assessed the case report and literature review by Raho et al., titled "Cerebral Venous Thrombosis during Thyrotoxicosis." The paper offers valuable insights into the association between thyrotoxicosis and cerebral venous thrombosis (CVT). The topic is of clinical relevance, and the manuscript presents a case study along with a review of relevant literature. The paper is well-structured and provides a clear overview of the case and its connection to CVT. The literature review is comprehensive and strengthens the case report by placing it in a broader context.
Suggestions:
- Discussion Focus: While the paper offers a comprehensive review of the literature, the discussion regarding the mechanisms linking thyrotoxicosis and CVT is somewhat brief. Expanding on this section with more in-depth analysis and possible hypotheses could strengthen the paper.
- More Critical Analysis: In the literature review, the paper mentions that in most cases, CVT is associated with additional known prothrombotic factors, suggesting a multifactorial genesis. This point needs more critical analysis and discussion regarding the implications for clinical practice.
- Visual Aids: Incorporating visual aids, such as tables or figures, to summarize key findings from the literature review could enhance the paper's readability and impact. Few figures quality is very poor.
- Clarity on the Case: Ensure that the clinical case presentation is clear and concise. This will help readers quickly understand the significance of this specific case in the context of the larger literature.
- Risk Factors: In the discussion, elaborate on the relevance of recognizing thyrotoxicosis as a potential risk factor, particularly in cases where no other procoagulative conditions are apparent.
I recommend this paper for major revisions. The manuscript has a strong foundation but would benefit from further development in the discussion section, as well as the inclusion of visual aids to enhance its impact.
This paper contributes to the understanding of CVT in the context of thyrotoxicosis, and with these suggested revisions, it has the potential to be a valuable resource for clinicians and researchers in the field.
Comments on the Quality of English LanguageMinor editing required
Author Response
Suggestions:
- Discussion Focus: While the paper offers a comprehensive review of the literature, the discussion regarding the mechanisms linking thyrotoxicosis and CVT is somewhat brief. Expanding on this section with more in-depth analysis and possible hypotheses could strengthen the paper.
Dear Reviewer, thank you for your insightful comment. As you indicated, we deepened this aspect in the Discussion section, please see:
“Specifically, a recent literature review examined the mechanism that might link the pro-coagulative state to altered thyroid function [76]. In particular, the importance of thyroid hormone levels on coagulative function has been demonstrated by the presence of significantly increased levels of vWF, fibrinogen, and D-dimer, even in patients with subclinical hyperthyroidism, when compared to euthyroid patients [77].”
“Furthermore, another study in healthy volunteers consistently showed that excess in thyroid hormones resulted in a hypofibrinolytic condition and an enhanced activated thrombin-activatable fibrinolysis inhibitor (TAFIa)-dependent prolongation of clot lysis [84].”
“Indeed, the seminal systematic review and meta-analysis by Stuijver et al. confirmed that hyperthyroidism shifts the haemostatic balance towards hypercoagulability and hypofibrinolysis in both endogenous and exogenous hyperthyroidism [13]. Coherently, another study reported that subjects with hyperthyroidism exhibited increased clot maximum absorbance in comparison to a control group, along with prolonged clot lysis time, and this was found to be positively associated with FT4 levels [85].”
“Therefore, to summarise, thyroid hormones act at different levels in determining hypercoagulability: they cause an increase in FVIII, in particular in its B subunit (FVIIIB), FIX, vWF, fibrinogen and PAI-1, acting directly on hepatocytes and endothelial cells, causing a state of hypercoagulability and hypofibrinolysis [11,82,86]. T4 levels, which are closely correlated with increased FVIIIB, FIX (natural inhibitor of activated protein C), SERPIN A5 and alpha2-antiplasmin, and the negative correlation of plasminogen (the precursor of fibrinolysis), thus increase the risk of thrombosis [13,83]. T4 also appears to act on the cytokine cascade, promoting the release of pro-inflammatory cytokines, including interleukin-1 (IL-1), which increases the production of ultra large vWF multimers, IL-6 and IL-8. IL-1 gene expression is also regulated by thyroid hormone homologues, including tetraiodothyroacetic acid, which, in turn, regulates transcription of the CX3CL1 chemokine gene [87,88]. The latter is released by endothelial cells in response to inflammatory factors and increases platelet adhesion [82]. Finally, T4 crosses the blood-brain barrier by binding with transthyretin (TTR) at the choroid plexuses, promoting cerebral and cerebrospinal fluid uptake of the hormone [89]. This uptake plays an important role in brain development, but, when excessive, also has negative effects due to its procoagulative action [90].”
- More Critical Analysis: In the literature review, the paper mentions that in most cases, CVT is associated with additional known prothrombotic factors, suggesting a multifactorial genesis. This point needs more critical analysis and discussion regarding the implications for clinical practice.
Thank you for your suggestion. We added some sentences about this in the Discussion section, please see:
“To summarise, in the diagnostic work-up of CVT, as for other conditions with multifactorial and complex etiopathogenesis, it is advisable in clinical practice to use a strategic and systematic approach, first searching for more probable causes of disease, later broadening the spectrum, looking for rarer but still plausible conditions. An appropriate diagnostic procedure should therefore include a search for genetic or acquired coagulopathies, autoimmune conditions, history of vaccination, possible neoplasms, systemic or local infections, history of trauma, use of oral contraceptives or hormone replacement therapy, conditions such as pregnancy and puerperium in women. In this context, given the growing evidence, screening for thyroid function should also be performed, particularly in patients with a history of other autoimmune diseases, female, young age, and also in individuals with no history of thyroid disease. Such an approach should help the clinician identify all possible underlying causes of a CVT in order to target the therapy and avoid any recurrence.”
- Visual Aids: Incorporating visual aids, such as tables or figures, to summarize key findings from the literature review could enhance the paper's readability and impact. Few figures quality is very poor.
Thank you, we have added some graphs summarising the main clinical and demographic characteristics of the patients described. We have also improved the resolution of some figures.
- Clarity on the Case: Ensure that the clinical case presentation is clear and concise. This will help readers quickly understand the significance of this specific case in the context of the larger literature.
Dear Reviewer, thank you for your suggestion. We modified. Please see the Case presentation section.
- Risk Factors: In the discussion, elaborate on the relevance of recognizing thyrotoxicosis as a potential risk factor, particularly in cases where no other procoagulative conditions are apparent.
Thank you for your advice. We added this in the Discussion section, please see:
“Thus, the risk of thrombosis has been reported in series of patients with hyperthyroidism. Thyrotoxicosis therefore should be recognized by the clinician as a possible underlying cause of CVT, particularly when other pro-thrombotic factors are not found. In contrast to several other risk factors for CVT, most thyroid disorders are easily and effectively treatable, and the detection of any thyroid dysfunction underlying CVT could lead to the latter being considered as "provoked" which shortens the duration of anticoagulant therapy. Therefore, thyroid dysfunction should be considered as a relevant risk factor for CVT in clinical practice, and it would be useful to evaluate thyroid parameters in all patients with CVT.”
Finally, we revised the language with the help of a native speaker expert.
Reviewer 3 Report
Comments and Suggestions for Authors
Comments:
1) CVT is often associated with both hypothyroidism and hyperthyroidism, from literature context, thyroid deregulation is the cause or the consequence of the CVT. Please discuss with suitable references.
2) Mention the related references in context with comment#1 for e.g. 10.1007/s00415-008-0746-5 (J Neurol. 2008;255(7):962-6.), add a discussion how hypothyroidism can be a risk factor and/or pro-coagulant.
3) Multiple case reports are already available in the literature showing association of CVT and thyrotoxicosis. How the present case report is unique and adding information to the available literature. A statement to be added in the conclusion section to support the comment.
Comments on the Quality of English LanguageMinor Typos editing required
for e.g. line 101, thyrotixicosis? (thyrotoxicosis)
Author Response
Comments:
1) CVT is often associated with both hypothyroidism and hyperthyroidism, from literature context, thyroid deregulation is the cause or the consequence of the CVT. Please discuss with suitable references.
Dear Reviewer, thank you for your valuable insights. We have described in more detail the mechanisms by which hypothyroidism and hyperthyroidism can cause CVT, please see the Discussion section. We have also added some sentences to explain how thyroid dysfunction could be a consequence of CVT, please see:
“Another interesting point might be to consider thyroid dysfunction as a consequence of CVT. Few cases have been reported in the literature of cavernous sinus thrombosis, usually bilateral, determining hypopituitarism. Often this is a septic thrombosis, with the extension of infection to the hypothalamic-pituitary site. The resulting disruption of the hypothalamic-pituitary-gland axis has been associated with multiple hormonal dysfunctions, including thyroid dysfunction [91-94].”
2) Mention the related references in context with comment#1 for e.g. 10.1007/s00415-008-0746-5 (J Neurol. 2008;255(7):962-6.), add a discussion how hypothyroidism can be a risk factor and/or pro-coagulant.
Dear Reviewer, thank you for your suggestion. We added a description about this in the Discussion section, please see:
“During hypothyroidism both hypercoagulability and hypocoagulability have been reported, depending on the degree of thyroid dysfunction, severe hypothyroidism being more frequently associated with hypocoagulability [62-64]. It seems to be associated with several dysfunctions in the coagulation cascade and in platelet adhesion and functioning, in particular: reduced platelet count, adhesion and aggregability; reduced values of vWF (i.e. acquired von Willebrand syndrome), FVIII, FIX, factor XI (FXI), reduced fibrinogen and increased fibrinolysis [65,66]. Moderate hypothyroidism, subclinical hypothyroidism, and autoimmune thyroid diseases, on the other hand, would appear to be more frequently associated with a pro-thrombotic state, due to increased fibrinogen, FVIII, and PAI-1 levels, increased mean platelet volume, decreased antithrombin-III and a global decrease in fibrinolytic activity [62,67-70]. Moreover, hypothyroidism also seems to contribute to slowing in venous flow with an indirect mechanism, leading to a left ventricular dysfunction and, consequently to a decrease venous velocity. Hypothyroidism-induced endothelial damage has already been reported for arterial vessels, as well as an increased prevalence of anti-endothelial cells antibodies in hypothyroid patients, suggesting a similar mechanism of damage on the venous side [71].”
3) Multiple case reports are already available in the literature showing association of CVT and thyrotoxicosis. How the present case report is unique and adding information to the available literature. A statement to be added in the conclusion section to support the comment.
Thank you for your advice. As you suggested we added a statement about this in the Conclusion section, please see:
“The available evidence in the literature regarding the association between CVT and thyrotoxicosis involves a few case reports and case series not quite recent. This paper has the advantage of collecting the different available evidence, comparing it with each other and with the case described, and discussing the possible implications regarding considering thyroid dysfunction as a possible risk factor for CVT, in order to gain resonance in clinical practice.”
Finally, we revised the language with the help of a native speaker expert.